# Linkage of Sustainability to Environmental Impact Assessment Using the Concept of Ecosystem Services: Lessons from Thailand

## Kanokporn Swangjang

Department of Environmental Science, Faculty of Science, Silpakorn University, 6 Rajamankha Nai Road, Amphoe Muang, Nakhon Pathom 73000, Thailand; swangjang_k@su.ac.th; Tel.: +66-34-245-330

**Abstract:** The concept of ecosystem services (ES) could help Environmental Impact Assessments (EIAs) contribute toward sustainability goals. This study aimed to systematically analyze the ES contents contained in Environmental Impact Statements (EISs) to ascertain whether they included appropriate data as a tool for project implementation in line with sustainability practices. The EISs were analyzed using the review criteria used to assess the criteria of good EIA practice, and these criteria were then integrated and linked to the concept of ES. The results indicated that the treatment of environmental impact studies from the perspective of impact assessment practice was advantageous; however, opposing results of the integration of ES in environmental impact studies were also found. The quality of EISs that reflect ES primarily depended on the project type. The highest quality of content to support ES was for baseline information. The contents in response to biodiversity and the relationship among sustainable indicators in the baseline stage, including the assessment of biodiversity, especially its loss and gain, and the identification of measures focusing on mitigation hierarchy, were inadequate. Consequently, these constraints affect the use of impact assessments as a tool to promote project activities in sustainable ways. An approach to integrate ES in EIAs was developed based on the findings of this study.

**Keywords:** environmental impact assessment; ecosystem services; environmental impact statements; content analysis; sustainability; Thailand



## 1. Introduction

Environmental Impact Assessment (EIA) has long been recognized as a process that provides sufficient mechanisms for regulating the environment and all development projects related to sustainable development [1]. Over the past five decades, EIA requirements have been adopted, in various forms, in planning, policy, and higher levels of legal hierarchy [2]. The development of EIAs is ongoing in more than 80 countries, in terms of laws, regulations, and implementation [1]. At the same time, EIA knowledge is diversifying, not just at the project level but also at strategic levels and across many disciplines [3]. EIAs help to ensure that environmental and socio-economic issues can be identified and addressed throughout the planning and implementation phases of projects and at the higher levels of decision making [2,4]. An EIA should provide sufficient information and justification to enable decisions to be made, based on predictions of the potential effects of a development, and identify ways to reduce and mitigate unacceptable impacts [5]. It is widely understood that the EIA is one of the main mechanisms for regulating the environment and all development projects related to sustainable development [1,6,7].

Ecosystem services (ES) can be defined as the benefits that humans obtain from ecosystems [8]. This concept is a valuable tool for transforming ecological knowledge into economic information, quantifying natural resource management, and enhancing the decision-making processes of developers [9,10]. According to Millennium Ecosystem Assessment (MEA) [8], ES can be grouped into four categories: supporting services are natural processes that maintain other ES categories [11]; provision services deliver ecosystem products [12]; regulating services control ecosystem processes, for example, through

biogeochemical cycles and biophysical structures at a variety of scales [13]; and cultural services are the intangible benefits that humans gain from nature [14]. These categories reinforce the understanding of both supply and demand in ecosystems and their carrying capacity. ES directly support the goals of sustainable development [4], as they include a wide variety of benefits for people from existing ecosystems, and they affect ecological sustainability, social equity, and economic efficiency [3]. Since 2007, interest in the theory of ES and its relevance to policy implementation has increased [12], and ES have become particularly important in the field of international politics [15,16].

Potschin et al. [9] noted that ES are used to support the relationship between environmental assessment, the ecosystem, and environmental monitoring. The linkages of ES to environmental assessment provide the structure in multi-purposes for sustainability, which includes environmental, social, and economic dimensions [17]. However, the adaptation of the ES approach is based on the aim of an intended study or proposed development [18]. Among these is the EIA.

In 1992, the United Nations (UN) Conference on Environmental Development (UNCED, Agenda 21: Principle 17) [19] indicated that:

*"Environmental Impact Assessment, as a national instrument, shall be undertaken for proposed activities that are likely to have a significant adverse impact on the environment and are subject to a decision of a competent national authority".*

Although the aim of an EIA is to support sustainable development during the implementation of a project, it remains imperfect [20]. One reason for this is that, at all stages of an EIA, it depends on legal enforcement in each country and region. The ES concept considers the relationship between the supply and the demand of ecosystems, which are affected by a project's sustainability. Thus, it involves not only the function of the ecosystem but also socio-economic development considerations. The ES concept could contribute to sustainability through changes in environmental and economic factors [21] and is closely related to people's ways of life [14]. In contrast, imbalances between ES supply and demand can show the unsustainability of a project's development. Thus, ES can support the aims of an EIA in line with its sustainability goals.

Research concerning the integration of ES in EIAs has received increasing attention over the past ten years [22–24]. Bouwma et al. [25] evaluated the adoption of the ES concept in twelve European Union (EU) policies that directly related to the use of natural resources and found there was a gradually increasing use of the ES concept in EU policies; however, there were differences in policies concerning environmental impacts [1]. Landsberg et al. [11] initiated a framework for incorporating ES in EIA that considered the interaction between the project and human well-being as the direct and indirect drivers of ecosystem change. Karjalainen et al. [26] incorporated ES into EIA by using multi-criteria decision analysis (MCDA) and found that the concept of ES could add value to the assessment process. Honrado et al. [27] developed a framework to analyze EIA practices and inferred the ES based on evidence provided by the EIA and Strategic Environmental Assessment (SEA) documents and other supporting information. Tallis et al. [28] provided an integrated framework for the improvement of biodiversity and the mitigation measures for ES that could improve the EIA method. A critical analysis of the potential role of ES in five case studies conducted by Baker et al. [29] provided a comprehensive approach, in which the ES were fully managed for the impact assessment framework. In the approach of Geneletti [3], the impact assessment was used as an essential tool to focus on spatial planning policies for future ES. Rosa and Sánchez [30] assessed impacts on ES and applied the ES approach to review the impact assessment of a mining project.

Early conclusions from more than 10 years of research into ES frameworks for EIA suggest that an ES approach could facilitate the identification of objectives and integration of environmental aspects [31]. The framework for ES in EIA has been introduced and developed on a case-by case basis, dependent on the nature of each region and the development of knowledge in this discipline. Much research into ES in EIAs has been conducted recently

in Europe, but such work is rare in Asia, especially in the Association of Southeast Asian Nations (ASEAN) countries.

The ES research that has been conducted in the Asian region has mostly been concerned with trade-off values of ES in sensitive areas. This is because most of these areas are located in the tropical zone, which is rich in the supply of natural capital. For example, Brander et al. [32] estimated the change in value of ES due to changes in mangrove forest cover in Southeast Asia during the period between 2000 and 2050 using a meta-analysis, while a rigorous economic estimation of mangrove ES in the Philippines was performed by Menéndez et al. [33]. Rasmussen et al. [34] assessed the actual use of provisioning ES and the ES availability in Laos PDR, focusing on agricultural land. Leimona et al. [35] provided empirical observations of trade-off mechanisms for ES in Indonesia, the Philippines, and Nepal. Kibria et al. [36] estimated the monetary and non-monetary value of ES in a protected forest in Cambodia. Intralawan et al. [37] examined the trade-offs between electricity generation and ES in the lower Mekong Basin. A study of ES indirectly related to the EIA context was performed by Shoyama et al. [38], who examined the approaches used in the evaluation of ES and found that modeling and biophysical indicators were the most commonly used methods, while the geographical distribution and the practical use of models remained limited. Abcede Jr and Gera [39] found that the inconsistencies and differences between different legal frameworks were the weak points in promoting ES for mining developments in the ASEAN region. They also highlighted critical issues with ES in ASEAN countries, including a prevailing lack of connections between ES legal frameworks among the member states, a lack of broad provision of ecosystem management and conservation, and a lack of coherent identification, targeting, and systematic integration of ES in environmental regulations in relation to mining.

Much research into ES in EIA has identified these limitations, but it has also led to the improvement and promotion of sustainable development in practice. This has provided the opportunity to integrate an ES approach in EIA, which has rarely occurred in the ASEAN region. The present study initiated the direct integration of ES with Environmental Impact Statements (EISs) as the product of EIA studies and used this as a tool for project control. This novel approach could connect the concept of ES to EIA as an effective project control mechanism, through a recognition of the balance of supply and demand in the area in which the project is located. The knowledge obtained from this study could help practitioners to integrate ES to ensure that they are addressed during project planning and development.

Checklist-based reviews have been an important tool to evaluate the effectiveness of EIAs [40]. The present study used a novel analytical approach to analyze the content of EIS, focusing on the ES perspective, for the following reasons. First, it is recognized that both EIA and ES can support sustainability. Their integration is valuable for promoting a sustainable approach to project-based EIAs. Second, the product of an EIA, the EISs, is a formal commitment to control a project's activities. If the content detailed in an EIS is based on the ES concept, it could ensure the quality of mitigation and monitoring measures as tools to control the actual completion of the project. That is, this approach aims to use ES as a valuable concept that could be effective in practice.

In Thailand, one of the ASEAN member states, both legal and organizational progress in EIAs is being made; however, the outcomes are yet to meet the target of sustainable development [20]. To support a sustainable target in EIAs, the integration of ES in EIAs is crucial. This led me to investigate the content of EISs in Thailand to ascertain whether they included appropriate data that could be used as a tool for sustainable project implementation. The aim was to integrate the approach to strengthen ES in environmental impact studies, which could represent both direct and indirect factors for the effectiveness of EIAs.

## 2. Methods

### 2.1. Case Study

In Thailand, the concept of ES is new, even though Thailand is geographically located in the tropical zone, where the ecosystem is complex and unique. ES were included in

Thailand's 11th National Economic and Social Development Plan 2012–2016 as part of the Sustainable Natural Resource and Environmental Management Strategy [41], but it has not been completely implemented. Consideration of ES has been stipulated since Thailand's 12th National Economic and Social Development Plan 2017–2021 to enhance the ecosystem approach and generate income from conservation, according to the principles of ES [42]. Although the use of EIAs in Thailand is very advanced, the promotion of ES for legal purposes is still at an early stage [20]. In contrast, EIA, as the approaches considered at project level, has been strengthened in all authority hierarchies. EIA in Thailand was officially initiated in 1975 as part of the National Environmental Quality Act (NEQA). It was prescribed in the Constitution of the Kingdom of Thailand in 1997, and since then has been changed over time until it was finalized in 2018. There are three levels of environmental impact study at project level (Table 1), namely Environmental Health Impact Assessment (EHIA), Environmental Impact Assessment (EIA), and Initial Environmental Examination (IEE). The type and size of assessments required for development projects depend on the perceived seriousness of their impacts. The details included in the studies differ from each other. EHIA documents are the most significant, both in terms of the details and the presentation of the project, while IEE documents include the fewest details.

**Table 1.** Comparison of IEE, EIA, and EHIA.

| Issues | IEE | EIA | EHIA |
|---|---|---|---|
| Legal enforcement | NEQA * since the 1992 Government Gazette | NEQA since 1975 | Constitution of the Kingdom, since 2007 |
| The number of projects required (as of 2022) | 2 projects [1] and 10 projects in protected areas [2] | 35 projects and 3 projects in protected areas | 11 projects and not allowed in protected areas |
| Project significance | Moderate impact | Moderate to high impact | The greatest impact |
| Public participation [3] (as of 2022) | One time During an EIA study | Two times During scoping and drafting of the final document | Three times During scoping, EIA study, and drafting of the final document |

**Sources:** [1] [43]; [2] [44]; [3] [45]. **\* NEQA:** Thailand's National Environmental Quality Act.

The selection of the EISs for this review was based on a subjective view. The selection criteria depended on judgments regarding a representative of each document type. For EIA, activities related to different types of projects were used, while for IEE, the consulting firms or approved projects were used. Therefore, condominium and housing projects, studied by different consultant companies, were selected as an example for IEE. An exploration and oil production project was selected for EIA because it had a major impact, and the project activities were different from those of a condominium project. For EHIA, the assessment year was used because the project activities for all EHIAs were classified as having highly important impacts. Different generations of EHIA over time illustrated the development of EHIA studies. The selected EISs are shown in Table 2.

**Table 2.** Details of the Environmental Impact Statements.

| EISs | Consulting Firm * | Project Type | Project Size | Year |
|---|---|---|---|---|
| IEE1 | A | Condominium | 32 rooms within a protected area | 2016 |
| IEE2 | B | Housing | 1.6 hectares | 2016 |
| EIA1 | C | Petroleum exploration | An area of 2-km radius | 2007 |
| EIA2 | B | Condominium | 70 rooms within a protected area | 2016 |
| EHIA1 | D | Petrochemical | Product expansion to 90,000 tons/year | 2012 |
| EHIA2 | D | Battery factory | >10 tons/day | 2016 |

\* For reasons of confidentiality, letters have been used to represent the various consultancy companies.

*2.2. Content Analysis*

A qualitative analysis of EISs in environmental impact studies was conducted, using content analysis to identify how ES are dealt with in EISs. Analysis of EISs has been conducted since the late 1980s [46]. A review checklist was developed from time to time, not only to ensure the completeness of EISs, but also to check the performance of EIA processes, based on their intended purposes. It was determined that an EIS itself is sufficient to provide the requisite information from developers, project designers, and operators to government authorities and the public about a proposed development [47].

Several studies have been developed that involved a review of EIS contents [48–52], and a review of the EIA guidelines has been conducted [47]. However, no consideration of EISs that reflected ES has been carried out for the ASEAN region, which is located in the tropical zone with an abundance of ES. Analysis of EISs should demonstrate whether the information included provides an adequate basis for considering ES in practice.

The criteria for the main aspects of ES in the environmental impact studies were: existing environment, impact assessment and mitigation measures, and monitoring measures. The concept of audit criteria was adopted and included two main aspects, namely, whether there were sufficient data for an EIA study and whether the criteria integrated the ES concept (Table 3). A variety of bibliographic sources, both as indicators for EIS analysis and the ES literature [3,9,11,12,20,26,28,53–61], were used.

**Table 3.** The criteria used for linkage of ES to EIA.

| Stage of an EIA Study | Criteria Topics | Codes for the Criteria |
|---|---|---|
| Baseline study | Sufficiency of baseline data | B1–6 |
| | Provide a framework for ES | B7–8, 13–14 |
| | Content support for ES | B9–16 |
| Impact assessment | Sufficiency for impact assessment | I1–5 |
| | Content support loss/gain in an ecosystem | I9–11 |
| | Content support for ES | I6–17 |
| | Linkage to ES compensation | I15–17 |
| Mitigation monitoring | Sufficiency for mitigation/monitoring | M1–3, 9–11 |
| | Provision of a linkage to ES | M4–5 |
| | Mitigation/monitoring support for ES | M6–8 |

These audit criteria were then classified using the codes for each category according to the content of the EISs. The extent to which EISs met these criteria was subjectively assessed using a five-point Likert scale [48] (Table 4). The average total score for each category was calculated to reflect the content of EISs that supported ES.

**Table 4.** Criteria indicating the level of consideration.

| Level | Criteria | Detailed Response |
|---|---|---|
| 5 | Complete | Provided complete information about issues related to the set criteria; no further supporting information necessary |
| 4 | Sufficient | Provided sufficient information, only minor information required for more completeness |
| 3 | Adequate | Provided details related to the set criteria; lacked some important information |
| 2 | Inadequate | Only provided general details with no responses to the set criteria |
| 1 | Deficient | Missing details in a particular category |

## 3. Results and Discussion

*3.1. Ecosystem Services in Environmental Impact Studies*

The criteria reflecting the quality of an environmental impact study in support of the ES were divided into three parts: the baseline description (project description and existing environment), impact assessment, and mitigation and monitoring measures, as follows.

### 3.1.1. Baseline Study

The first stage of an environmental impact study is the baseline study (Figure 1). The scores of contents of the EHIAs were higher than those of the IEEs. Criteria reflecting good EIA practice, following Sadler et al. [61], were initially indicated (B1–B6). For all EISs, the data presentation was scientifically well defined, but the boundary of the area for environmental impact studies (B2) was not flexible. A defined distance of either 1 or 5 km from the project location was usually used. The types of environmental components considered in baseline studies relied upon formal guidelines and not the specific characteristics of the area. Alternative identification (B1), land use (B4), and urban planning (B5) were satisfactory; however, the scores of characteristics of the area where a project was located that related to the balance between conservation and development (B3) were lower, with most at exceptional levels. This is an important issue for the linkage to ES of the areas.

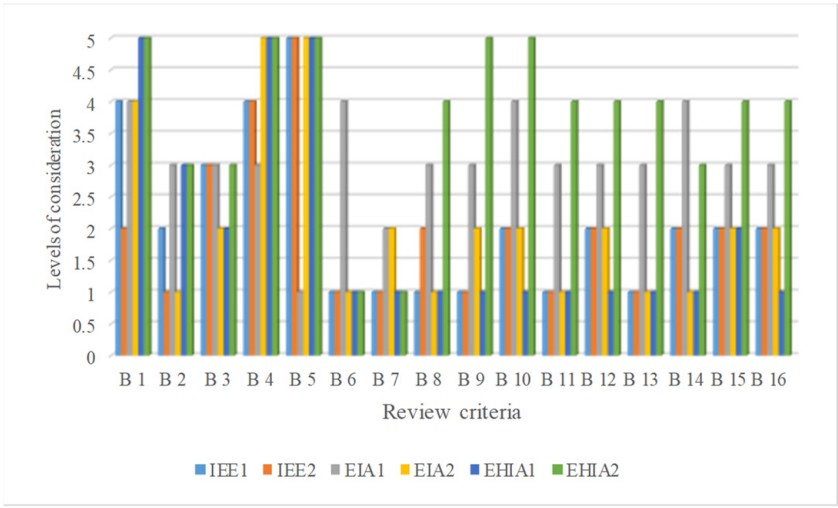

**Figure 1.** Values of ES in baseline study of EISs content.

For the integration of the ES concept in baseline studies (B7–B16), a good ecological study is important for the outcome to support ES, as it can provide the necessary framework, not only to obtain an effective EIA [62] but also to provide spatially explicit data for ES [63], as a way of linking ecosystem stocks to defined ES of an area to support human benefits [64]. The scores of data supporting ES (B7–B16) were lower than the basic concept of EIA (B1–B6). The consideration of project location areas in terms of their sensitivity (B12) and their features (B13) for biodiversity were satisfied for high-impact projects, especially at the EHIA level. These criteria could link to habitat-loss considerations and the functional connectivity among ecosystems across landscapes, which is important for strengthening the biodiversity of habitats and could be adapted to sustain ES values in the subsequent stages of impact assessment.

The linkage between the ecological, social, and economic data based on land-use considerations (B8) was an advantage for EIA 1 (score 3) and EHIA 2 (score 4). These baseline data were sufficient to consider the balance between the supply of resources and the demands made by a project. However, they were not used for further intensive assessment of an ES approach. Integration of environmental and socio-economic contexts (B7), which could promote the ES approach in EIA [30] and provided the baseline for ES payments for further assessment [35], was missing for the IEE and EHIA groups.

The recognition of the importance of legal encouragement was found in EIA 1 and EHIA 2, in which both national legislation and international agreements related to biodiversity and sustainable development were reviewed and highlighted (B11, B14). The obligation to conduct EIAs could confirm their effectiveness [40] during the baseline stage. Overall, the results showed that the basic information to support ES values in the EISs was most evident for EIA 1 and EHIA 2.

B1. To specify likely project alternatives
B2. To specify a flexible boundary area
B3. To consider the characteristics of the area and the appropriateness of conservation and development
B4. To consider land use
B5. To consider urban planning
B6. To consider international agreements
B7. To identify reasons to support the balance between environment and socio-economic indicators
B8. To consider the linkage between ecological and socio-economic factors
B9. To identify a specific ecological boundary
B10. To identify ecosystem types
B11. To present and review legislation related to biodiversity
B12. To consider a sensitive area for biodiversity
B13. To visit the study area based on the features of its biodiversity
B14. To detail the laws and regulations that contribute to sustainable development
B15. To integrate ecological, social, and economic data based on land use consideration
B16. To provide priority of supply and demand on the basis of baseline information

### 3.1.2. Impact Assessment

In the following section, the EIA (Figure 2) components presented in the baseline data (project description and existing environment sections) could be used to evaluate impacts or be combined with other environmental components in impact assessments of a specific environmental component. For example, at the EHIA level, air quality, terrain, and land use were combined to evaluate impacts on air quality. In contrast, it was found that many components were present only at the baseline stage and not used further for impact assessment. The criteria reflecting good EIA principles [61] were I1 to I5, with the highest average score being for EIA 1. In this EIS, the consistency of the causes of the impacts that affected the sustainable components was considered based on biodiversity within the proposed area (I3). The achievement dependes on the understanding of the ecological baseline of the areas. This criterion (I3) could connect EIA studies to the ES approach, in which the relationship between project demands and ecological values of an area is considered. A lack of alternative evaluations of the project (I2) was the weak point for all EISs, even if these criteria had satisfactory assessment scores in the baseline study stage. In contrast, the best average score for the impact assessment was for the consideration of a project's life cycle and the members of public who were affected (I5) (score 3.00). The public interest is crucial to incorporate ES in EIAs [30].

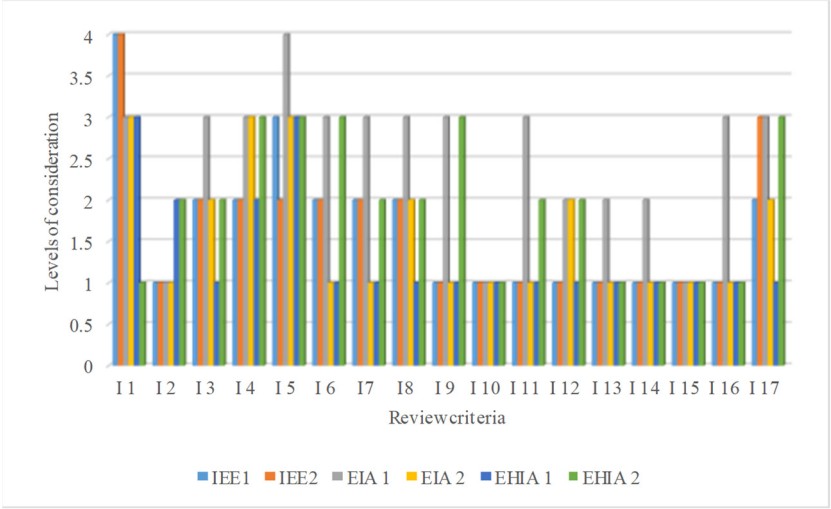

**Figure 2.** Value of ES in impact assessment of the content of EISs.

Although mathematical models were used for some components, in particular, air quality, quantitative details were used only for the values of specific parameters, without further connecting their effects with the supply and demand in the ecosystem. Due to the failure to incorporate indirect effects in impact assessments, the connection to the ecological impact assessment (I6–I7) was at a low level. These details were, for example, *"species of organisms are so common hence a low impact is predicted"* or *"... the project's wastewater was collected in the combined wastewater treatment system, so the "impact on bi-ological resources is negligible".*

The criteria corresponding to support for ES were I9–I17, whereas I4 to I12 provide details about the potential carrying capacity of an area [65]. I15 and I16 provided opportunities to support good mitigation and monitoring in the subsequent stages of environmental impact studies [66]. I17 provided the linkage between ES to the sustainability of a project.

Data retrieved from baseline studies affect the uncertainty of impact assessments. Qualitative assessments depend on the input baseline data [31], which is generally subjective. Qualitative assessments (I9) in the EISs reviewed were used more than a quantitative approach (I10). According to Karlson et al. [31], a quantitative ecological approach could also be used to relate ecological processes to the wider context and thereby reduce uncertainties of the predicted effects outside of a project corridor, together with long-term effects. A quantitative approach to biodiversity could also provide an understanding of the combined effects of habitat loss, species population decline, and the loss of other ecosystem mechanisms [64]. Unfortunately, a quantitative assessment of biodiversity was missing from all EISs. These aspects could support ES if they were integrated in an assessment of trade-off values in the ecosystem where the project is located.

The assessment of impacts on a single component did not accurately reflect the benefits of supply and demand in an ecosystem. The incorporation of the "no net loss" and "net gain" concepts of biodiversity (I11), together with the other biodiversity criteria, were mostly missing in considerations of impact assessment, due to the failure to conduct further assessments of indirect impacts, especially residual and cumulative impacts (I15). Overall, the consistency of ecological impact assessment based on project features (I17) was at a moderate level, especially for project IEE and EIA levels.

---

I1. To assess the impact covered by a project's life cycle
I2. To analyze project alternatives
I3. To assess impacts based on an ecological baseline
I4. To consider other sustainable components
I5. To cover all affected members of the public
I6. To clarify ecological impact identification
I7. To clarify ecological impact evaluation
I8. To analyze the ecological impact, focusing on the risk to ecosystems
I9. To assess the impact on qualitative biodiversity
I10. To assess the impact on quantitative biodiversity
I11. To assess the impact on the loss or gain of biodiversity
I12. To consider impact severity based on the sensitivity of biodiversity
I13. To consider impact severity based on the resilience of biodiversity
I14. To consider impact severity based on the recovery of biodiversity
I15. To assess residual and/or cumulative impacts
I16. To arrange the impact hierarchy on biodiversity
I17. To assess ecological aspects consistent with the project characteristics

---

### 3.1.3. Mitigation and Monitoring Measures

The criteria used to reflect EIA practices were explored (M1–M3) (Figure 3). The criteria used to identify mitigation and monitoring of the results of impact assessments (M2) were outstanding in all EISs. The best score was once again for EIA 1, which was a well-defined requirement for project control activities. This issue is important because an impact assessment's output should be used as a tool for project implementation. Mitigation

and monitoring identification in EHIAs was better than in IEEs. An alternative aspect (M3) was lacking in most of the EISs reviewed. This issue is important for the strategic level, especially SEA, which is known to be higher level recognized as a sustainable tool in the EIA approach [29].

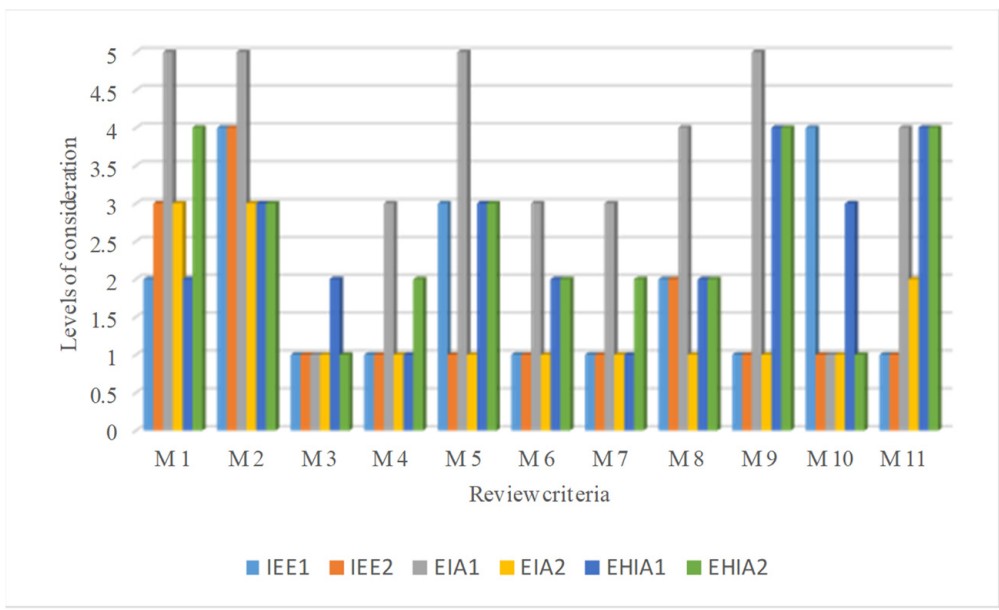

**Figure 3.** Values of ES in mitigation and monitoring measures of EISs content.

It is important to ensure that ES in EIA commitments are reflected in practice. The criteria that directly reflected ES, including mitigation of biodiversity losses (M4) and compensation for the loss of biodiversity (M6 and M8), were apparent in EIA and EHIA projects; however, they focused on compensation plans for the area, without any linkage to ES trade-offs. Mitigation hierarchy (M7), which has been recognized as an essential tool for project control mechanisms [67] and could link an EIA to ES [68], was sufficient for the EIA petroleum exploration project (EIA1). The lack of mitigation hierarchy consideration is a key recurring problem of environmental assessment practice [69]. Consequently, biodiversity offsets, which are directly related to the supply within ecosystem (M8), were lacking for IEEs and unclear for EHIAs. The best score was for EIA 1, in which the consideration of biodiversity loss and programs to control it were specified and covered in the project's life cycle.

Mitigation and monitoring could help to maintain the supply of priority ES [30]. The identification of opportunities to enhance or change mitigation (M9) and monitoring (M10) measures in cases of future findings of unpredictable impacts were crucial for EIA1 and IEE 1, respectively; however, these issues were not related to the level of program achievement (M11), which pays more attention to EIA practice than the achievement of ES trade-offs.

M1. To identify the measures based on survey data and public opinion
M2. To identify the measures agreed with the result of the impact assessment
M3. To consider alternative measures
M4. To identify mitigation of biodiversity losses
M5. To consider the residual impact
M6. To identify the compensation for the loss of biodiversity at the species to the ecosystem levels
M7. To consider the mitigation hierarchy
M8. To establish a compensation plan for ecosystems
M9. To provide an opportunity to enhance or change the mitigation measures
M10. To provide an opportunity to enhance or change the monitoring programs
M11. To consider the achievements of the measures

The scores for the quality of the baseline information in the different types of EISs ranged from 2.00 to 3.75. Surprisingly, the quality of the information for impact assessments, which is a crucial stage, was found to have the lowest average score (1.38). For mitigation and monitoring, the average score was 2.24. The project type influenced the quality of the mitigation and monitoring, as the scores for EHIAs were higher than those for IEEs (Figure 4).

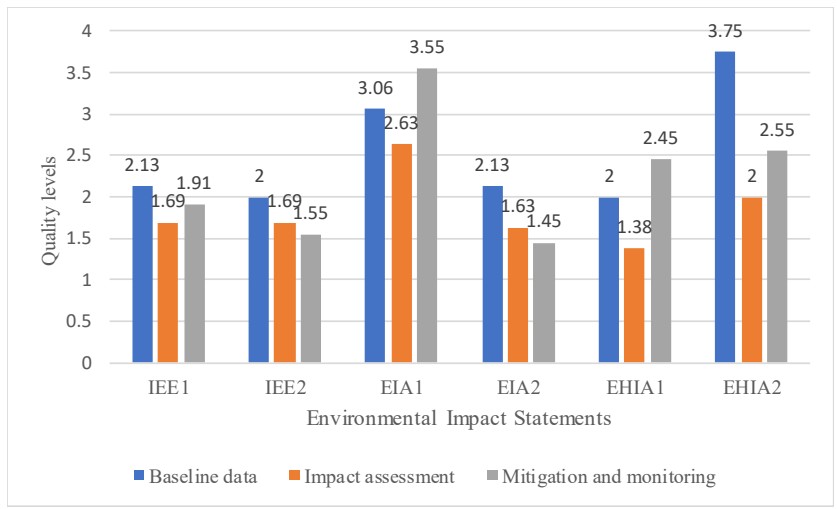

**Figure 4.** ES quality levels in Environmental Impact Statements.

### 3.2. Integration of Ecosystem Services in Environmental Impact Studies

The findings described in Section 3.1 provided an opportunity to integrate ES in EIAs. The role played by ES can improve the understanding of the ecosystem mechanisms resulting from a project's activities. According to Karjalainen et al. [26], ES can be considered during the early stages of an EIA study through mitigation and monitoring. The possibility of ES integration in EIAs was reflected by the limitations and opportunities detailed in Table 5.

The findings of the present study outline how ES could be incorporated in environmental impact studies, as follows. First, in the scoping phase, the selection of appropriate sustainability indicators should be focused on and assessed throughout the EIA study phase. However, the adequacy of the biodiversity baseline and its links with the other components is important for providing satisfactory information for the services demanded from and supplied by an ecosystem [69]. Project activities should be the main consideration when determining the demand for natural resources, whereas the biodiversity component, from species to ecosystem, serves as the supply component within a specified area. Biodiversity is an important contributor to ES, which promotes livelihoods and well-being [70]. In EIA studies, biodiversity pertains to ecological studies as a primary component that could support a project's development in accordance with sustainable approaches [5,70–72]. The quality of the ecological content is crucial. Levels of biodiversity change will vary throughout the duration of a project. Therefore, these considerations must be accounted for in the EIA methodology [55]. The flexibility of the impact boundary is related to the nature of the study area. Basic criteria, including factors related to survey planning, flexibility in relation to the size of the proposed areas, the methods applied to explore specific ecological groups, and the initial site visit, are all important in determining the quality of baseline supply and demand data for an area.

**Table 5.** Integration of ES in EIA studies: Opportunities and barriers.

| Opportunities | Barriers | Approach to Integrate ES in EIA Studies |
|---|---|---|
| *Baseline description(project description and existing environment)* | | |
| - The presentation had a scientific basis.<br>- All projects complied with the enforcement of urban planning.<br>- Project alternatives and reasonable choices were identified.<br>- Ecological, economic, and social linkages were considered through the characteristics of land use, especially for EIA1 and EHIA2.<br>- The information linked to the ES included urban planning, land-use features, and the relationship between project activities and their ecosystem. | - The boundary of the study was fixed, lacking the flexibility to take the surroundings into account.<br>- The determining component depended on the official guidance rather than the project activity and the nature of the area.<br>- The presentation of ecological information was lacking, with only superficial information, especially for IEE projects.<br>- Survey and data collection were only performed once. | - Project activities and biodiversity components based on land-use conditions are the main supply and demand considered for ecosystem capacity.<br>- The environmental, social, and economic components should be considered equally, depending on the nature of the proposed area.<br>- Spatial and temporal coverage should be specified<br>- The biodiversity context is based on the quality of ecological study in the EIA. |
| *Impact assessment* | | |
| - Impact assessment covered project life cycle.<br>- The considerations of the affected public were satisfied. | - Project alternatives were not further assessed for their impacts.<br>- The weaknesses of ecological impact assessment were notable, especially the linkage to sustainable development.<br>- Impact assessment of indirect effects, together with residual and cumulative impacts, failed.<br>- The impact assessment was ambiguous, except for EIA1. | - Baseline data should provide the main issues to determine the assessment of impacts on service delivery and values of ecosystems.<br>- Criteria for impact identification and evaluation should focus on biodiversity values and cover both direct and indirect impacts, together with residual and cumulative impacts.<br>- The spatial extent of impact assessment should be considered as well as the biodiversity and associated consequent mechanisms, depending on the ES of the area. |
| *Mitigation and monitoring measures* | | |
| - The public's comments were included in the mitigation and monitoring identification.<br>- The quality of mitigation and monitoring was based on project levels related to the enforcement by competent agencies. | - Mitigation implementation was questionable, especially for IEE.<br>- Mitigation hierarchy was not well defined.<br>- Biodiversity compensation of net loss was not clear.<br>- Alternative mitigation and monitoring were lacking.<br>- The time span of monitoring and future modifications was lacking. | - Mitigation hierarchy should be highlighted. Avoidance of impacts is initiated, with subsequent hierarchy through biodiversity offsets.<br>- Residual and cumulative impacts are recognized for future mitigation enhancement.<br>- Appropriate biodiversity offsets should be designed in accordance with the landscape of the area.<br>- Monitoring focuses on high efficiency for project implementation. |

The strength of the baseline data was based on scientific knowledge. The linkage of ecology and socio-economic aspects to land use under the enforcement carried out by urban planning department was outstanding, and these relationships could support the carrying

capacity of such areas. The creation of new habitat by project replacement programs provided both an opportunity and a barrier to enhancing biodiversity in a landscape [73]. Fürst et al. [13] indicated the importance of land-use management as a factor supported by the regulating services concept. The flexibility of the biodiversity boundary was not included in the EISs evaluated. Consequently, these failures meant that assessments could not respond to the actual supply and demand within an ecosystem because the impacts on biodiversity were mostly indirect impacts resulting from physical impacts, such as air or water impacts. Moreover, existing quantitative studies, which consider changes at both the temporal and spatial scales of ecosystems, are critical for the successful integration of ES and environmental impact studies [74,75]. The temporal scale, which affects both the baseline data and the impact assessment, should include the characteristics of species that occur and an interpretation of the impact predictions [76]. Specification of the condition of the fauna and flora within an ecosystem is one approach that could support temporal coverage. This is vital in subsequent stages of an impact assessment. The lack of temporal data is problematic, especially for the small projects that characterized the IEE group, in which baseline data were only collected on a single date.

For an impact assessment, the project phases should include the project life cycle, at least in the construction and operational phases, and determine the resource demands. Losses and gains in biodiversity resulting from project activities should be the first priority for any impact assessment. According to Geneletti [59], an impact assessment should be based on ES as the priority, but legal guidance is needed that could indirectly encourage the assessors to conform. Baseline details should be further assessed for their impacts. Project alternatives, such as the project size, location, or processes, presented in the baseline data, should assess any impacts and be considered further for mitigation and monitoring. A weakness found by this study was the lack of alternative considerations throughout all stages of environmental impact studies. Remarkably, the impacts on ecosystems, from varying levels of human disturbance, were only addressed in qualitative terms. This raised some uncertainty regarding the overall impact assessments, which had been carried out without consideration of biodiversity concepts such as species loss, project effects on the natural habitat, and community and ecosystem components. Ecological impact identification and evaluation and contributions to environmental change can be negative and/or positive, residual and/or cumulative, and significant and/or magnitude impacts. Impact projections affect subsequent project activities. For example, negative impacts should be considered as a priority for effectively managing the reduction of adverse ecosystem effects.

Finally, mitigation and monitoring measures should consider ways to maintain supply and demand in ecosystems with reasonable costs and benefits. Options for mitigation, through the mitigation hierarchy, can help to improve and maintain the well-being of affected beneficiaries of ES [11]. A mitigation hierarchy is fundamental to EIA practice for biodiversity offsets, through the consideration of alternatives during program identification [77]. In practice, project compensation should adhere to the mitigation sequence of avoid, minimize, rectify, reduce, and compensate or offset [78]. These measures can compensate for the loss of ES resulting from a project's demands and is a crucial approach for incorporating ES in EIAs.

The benefit of mitigation and monitoring was the inclusion of public opinion, in accordance with legal enforcement. The responses should reflect the actual requirement of the local public [79]. To improve implementation, the members of the public identified should be representative of the public concerned, and impacts on poverty in both income and well-being should be considered. According to the UN Sustainable Development Goals, adopted in 2015 [80], poverty (Goal 1) also covers the poor in situations such as climate-related events and other natural shocks and disasters. Hence, all people should be considered equally.

A program's achievements are a vital element of a program's performance, and they should be exhibited by the project control agencies during project implementation. In this

regard, essential monitoring directly affects the likely implementation of EIA in practice. The cost of monitoring implementation is a fundamental principle that touches upon all monitoring-related activities. The ways to improve performance relate to these aspects. First, flexible programs are required, which can be achieved by optimizing the design. Second, the period during which parameters should be monitored should be included in a program's design. Third, the efficiency of resource use in monitoring should be a focus. The consequences of paying appropriate attention to these factors would be a reduction in the costs of monitoring unimportant parameters, which create a waste of effort. These savings could lead to better environmental management.

The measure of success of mitigation and monitoring is compliance. The effectiveness of mitigation and monitoring programs, all of which are defined in official documents, is not a guarantee that the programs will be implemented. This was confirmed by the findings of the present study in relation to the EISs of real estate projects. The mitigation and monitoring actions may have made it difficult to measure the accuracy of impact prediction, as the results of the impact assessment were not the key element when identifying the programs. Consequently, the monitoring of performance during project implementation was questionable.

Roe and Geneletti [81] noted that "*biodiversity underpins the delivery of essential ES on which the whole of humanity is dependent*".

However, this depends on the nature of the project and the environment where the project is located. To better incorporate ES in environmental impact studies, through eco-based objectives, close connections between biodiversity content and the different stages of an EIA are strongly recommended. These points can help an EIA to support the Sustainable Development Goals.

The integration of ES in EIA studies shown in Table 5 could support EIA guidelines, which are the formal EIA scoping stage for environmental impact studies. With regard to Thailand's EIA guidelines, ecological aspects adopted in these guidelines were inadequate and did not include linkages to the loss and gain relationships within the ecosystem in which a project is located [47]. For effective implementation, the stages of EIA studies require an understanding of ES concept by the assessors, hence, the promotion by a competent agency is necessary. During the follow-up stages, the connection of EIA projects to ES is part of the mitigation hierarchy that involves the protection and restoration of biodiversity values. The commitment to mitigation and follow-up should be rigorously enforced. Importantly, the ES concept should be directly incorporated in EIA legislation as the highest level of the legal hierarchy. The concept should focus on the implementation processes at a national level.

In the ASEAN region, EIAs have been clearly stated in Article 14 of the ASEAN agreement on the Conservation of Nature and the Natural Environment since 1985 [82]. In 2016, the ASEAN Socio-Cultural Community (ASCC) Blueprint Vision 2025 [83] was issued to promote the balance between societal and sustainable development in the ASEAN region. One of the objectives of this blueprint is to promote social development and environmental protection through effective mechanisms to meet the current and future needs of the people of the region. The impact of development projects on transboundary issues and the sustainable management of ecosystems and natural resources are key areas to be adopted. The findings of the present study could support the connection of ES to EIA for the purposes of sustainable development projects.

## 4. Conclusions

The findings presented here provide some insights regarding the integration of ES in EIA. These could be sequenced in each stage of EIA studies. Baseline data, the first stage, were found to have the highest quality content. This quality was dependent on the criteria of good EIA practice, which is the formal method used for EIA studies. The balance between environment and socio-economic factors, which provided the major support for ES during the following stages, was the weakest. As for the next stage, impact assessment, the

assessment of impacts on biodiversity was ignored, especially in terms of losses and gains of biodiversity resulting from impact activities. Although impact assessment followed the formal EIA guidance, extensive details to consider the balance between project demands and the supply from the ecosystem where a project was located were insufficient. This further affected the mitigation and monitoring measures in the final stage. The importance of mitigation hierarchy was not recognized, especially for the IEE level. All of these factors can affect the tool for project implementation. Hence, it is important to adopt ES aspects in the formal guidance for the enforcement of EIA studies.

Three aspects emerged from the content analysis. First, the quality of EISs, which reflect the ES, depended primarily on the project type, not the project group (IEE, EIA, and EHIA). In this regard, the legislation of the respective agency is paramount as it directly controls project performance. Second, the quality of EISs was sufficient for the formal process of EIA studies; however, the limitations for integrating ES in the EIA were considerable. Although good ecological baselines were detailed, the coverage impact assessment of biodiversity was weak during the subsequent stages. The weakness of ecological impact analysis, from the species level to biodiversity, directly concerns biodiversity compensation, which links the approach of environmental analysis to trade-off values in ecosystems. Finally, mitigation hierarchy as a tool to connect ES during the actual implementation was insufficient. Both mitigation and monitoring are warnings of the ecosystem stock that could provide the goods and services for a proposed project. Problems with the treatment of the content of EISs could be grouped into three limitations of the linkage of ES in EIAs: project proponents, ecological recognition, and the understanding of ES values by an assessor.

The evidence presented here also provides insights into aspects that hitherto have been overlooked. To extend the opportunities for future research, two main recommendations should be highlighted. First, regarding the analysis of EISs, as the impacts from project development can occur on both minor and major scales. The expansion to consider the major scale of project development is important, especially for large projects located in sensitive areas. A focus on the criteria regarding ecological aspects should be considered, as the initial stage of EIA study through mitigation and monitoring measures to control project implementation. Second, from the perspective of ES integration, the assumption of a trade-off between project demand and supply in the ecosystem in which a project is located was suggested. The recognition of trade-offs between different ecosystem services is needed, as it is not always the same ES that are being demanded by the project or being affected by the project.

According to Goal 15 of the UN Sustainable Development Goals (2015) [80], the integration of ecosystem and biodiversity values into planning and development processes should be promoted. The results of this study confirmed that the consideration of ES in environmental impact studies can be used to link the EIA to a project's sustainability. Any project located in the environment demands services from the ecosystem, while the supply is limited by the impacts of the project activities on the ecosystem. The concept of ES could promote a sustainable development approach in EIAs, in accordance with the Rio Declaration (1992) [19] and the ASEAN Agreement (ASEAN, 2017) [84].

**Funding:** This research was funded by the Office of the National Research Council of Thailand, grant Number SURDI 590155.

**Institutional Review Board Statement:** Not applicable.

**Informed Consent Statement:** Not applicable.

**Data Availability Statement:** Not applicable.

**Acknowledgments:** This document forms part of the main research on *environmental assessment based on sustainable development*. The author would like to thank the Department of Environmental Sciences, Faculty of Science, Silpakorn University, Thailand, for all the materials and support provided for this research.

**Conflicts of Interest:** The author declares no conflict of interest.

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
