# Peer review of "Linkage of Sustainability to Environmental Impact Assessment Using the Concept of Ecosystem Services: Lessons from Thailand"

_sustainability, doi:10.3390/su14095487_

Round 1
Reviewer 1 Report
Dear Author(s),
This paper is a well organized review on Environmental Impact Assesment and ecosystem services relationship in Thailand scale. In reviews, we would like to see key knowladges in a short discriptive paper to evaluate the ES and EIA connection. When I read paper, I have never bored. This is a critial point for a review assessment. In my opinion this will contribute the scientific literature about ES and EIA.
Reviewer 2 Report
The manuscript examined the environmental assessment contents of environmental impact statements to ascertain whether they addressed appropriate data used as a tool for sustainable project implementation. Overall, the approach of the study is good and could be useful in the public domain, but the manuscript needs considerable revision to reach the public domain. Authors are suggested to address following comments in order to make the manuscript suitable for publication.
* Abstract should be rewritten by detailing the aim and concept of the study. The abstract should state briefly the purpose of the study, the principal results and major conclusions.
*Provide significant words which are more relevant to the work in logical sequence as ‘keywords’.
* Introduction is very general and need to be elaborative to explore the actual philosophy to design the experiment. The introduction is insufficient to provide the state of the art in the topic. The originality and novelty of the paper need to be further clarified. What progress against the most recent state-of-the-art similar studies was made in this study?
*The introduction of the paper must be extended and reformulated in order to provide a more comprehensive approach.
*The manuscript does not provide interesting and technically sound discussion; it would be better to use more recent references in discussion.
*Authors have presented their result but these results obtained, authors did not justify in the discussion.
*Under section, discussion, it is recommended to discuss and explain what should be the appropriate policies based on the findings of this case study. Also, the results should be further elaborated to show how they could be used for real applications.
*Authors are suggested to draw major inferences/primary conclusions first quoting the data/results obtained followed by the secondary conclusions/ recommendations reached through the critical analysis/ investigation of the study. Based on the outcome of the study, the author(s) may recommend the extension of the present study as the future scope of research.
Round 2
Reviewer 2 Report
The author addressed all of my queries, therefore the manuscript may be accepted in its current form.